# Achieving Better Utility beyond LDP-FL by Independent Two-phase Protection

## Abstract

The Local Differential Privacy Federated Learning (LDP-FL) framework provides privacy protection by injecting noise at the client level. However, the noise accumulates in the model through the two-phase indivisible sequential process of LDP, thereby bringing the well-recognized privacy-utility trade-off challenge. In this paper, we propose an ideal interaction mode, Ideal Differential Privacy Federated Learning (IDP-FL), which allows for independent protection in the uplink and downlink phases. Through a comparative analysis of noise accumulation in IDP-FL and LDP-FL, we discover and theoretically prove that LDP-FL suffers from inherent *noise redundancy*, i.e. noise accumulation in uplink exceeds privacy requirements in downlink. To avoid this defect, we propose a novel framework, Noise Annihilation Differential Privacy Federated Learning (NADP-FL), which can be regarded as an instantiation of IDP-FL. In this framework, a portion of noises are distributedly generated in pairs, thereby mutually canceling each other out during aggregation and not appearing in the downlink phase. As a result, NADP realizes independent protection for both phases, eliminating unnecessary noise accumulation, achieving a more favorable privacy-utility trade-off and enhance protection in a way that incurs no further utility loss. We validate the superior utility, scalability and robustness of our framework through extensive experiments.

## 1 Introduction

With the rapid expansion of distributed databases and the ever-growing volume of data across domains such as smart homes (Li et al., 2023), transportation (Tahaei et al., 2020), and healthcare (Tang et al., 2019), ensuring secure and reliable data mining in decentralized environments has become increasingly critical. Federated Learning (FL) (McMahan et al., 2017) has emerged as a promising paradigm for privacy-preserving machine learning by allowing clients to collaboratively train models without exposing raw data. However, despite its decentralized nature, FL typically relies on a central server for model aggregation and distribution. In practice, assuming this server to be fully honest and trustworthy is often unrealistic—such trusted third parties are rare, and their compromise can lead to significant privacy breaches. Furthermore, FL remains vulnerable to privacy threats such as model inversion (Zhu et al., 2019) and membership inference attacks (Shokri et al., 2017), which can reveal sensitive information from seemingly innocuous model updates.

To address these threats, Local Differential Privacy (LDP) (Wei et al., 2020; 2021) has been widely integrated into FL by injecting noise directly on the client side. This ensures that individual data remains protected even when the central server is untrusted. Nevertheless, LDP introduces a fundamental challenge: the noise injected in the early stage (uplink) persists throughout the FL process and irreversibly propagates into the global model during aggregation. This results in a well-known trade-off between privacy and utility (Kim et al., 2021; Zhang et al., 2023).

In particular, the two-phase interaction in LDP-FL involves: (1) clients uploading perturbed local updates (uplink), and then (2) receiving the aggregated model from the server (downlink). Since noise is added only during uplink, it simultaneously protects downlink phase by propagating through aggregation. It is worth noting that the two phases remain tightly coupled, the amount of noise in each phase is inherently determined by the aggregation structure, rather than the actual privacy needs of each phase. Consequently, we are naturally led to ask: **Does this coupling lead to a mismatch**

**between the minimal noise required for each phase, resulting in excessive perturbation and degraded model performance?**

In this work, we answer this question positively. We identify and formalize an inherent flaw in LDP, which we term **noise redundancy**. This redundancy arises from the structural dependency between the two phases in LDP-FL: the noise injected in the uplink often exceeds the privacy requirement for the downlink, leading to inefficient noise allocation and unnecessary utility loss. To the best of our knowledge, this is the first work to formally define and analyze this problem.

To resolve this issue, we propose a new DP-FL interaction mode that decouples the uplink and downlink phases, and build upon it a novel framework, Noise Annihilation Differential Privacy Federated Learning (NADP-FL). Unlike traditional LDP, NADP enables independent privacy protection for both uplink and downlink phases. It introduced a part of structured noise pairs that cancel out each other during aggregation, effectively eliminating unnecessary perturbation. As a result, NADP can avoid noise redundancy and enhance protection in a way that incurs no further utility loss, with both theoretical privacy guarantees and experimental results showing superior privacy-utility trade-offs. Our contributions are as follows:

- We revisit the two-phase process of LDP (uplink and downlink) and theoretically prove that the LDP framework inevitably suffers from noise redundancy. To the best of our knowledge, this systemic defect in the LDP framework is raised for the first time.

- We construct a novel framework, NADP-FL, to avoid noise redundancy. Additionally, we theoretically prove that this framework satisfies $(\epsilon, \delta)$-DP and can enhance protection in a way that incurs no further utility loss.

- We conduct extensive experiments to validate our framework. It shows that, compared to LDP, NADP exhibits a better privacy-utility trade-off, offers better scalability, and inherently maintains a certain degree of robustness in dropout scenarios.

## 2 RELATED WORK

To mitigate the utility loss of LDP-FL, various strategies have been explored. Adaptive gradient clipping (Andrew et al., 2021; Fu et al., 2022) techniques dynamically adjust clipping thresholds based on data distribution to better fit different clients and training rounds. Dynamic noise scaling (Phan et al., 2017; Talaei & Izadi, 2024) allocates privacy budgets adaptively across training rounds, calibrating noise to changing model sensitivity. Per-layer budgeting (Errounda & Liu, 2023; Chen et al., 2023) approaches assign differentiated noise weights to model components based on their empirical contributions. More recently, shuffle-based methods (Balle et al., 2019; Chen et al., 2024) have been proposed to amplify privacy through trusted intermediaries that anonymize clients.

However, these approaches suffer from fundamental limitations: (1) Heuristic-based methods (adaptive clipping, noise scaling, per-layer budgeting) rely on task-specific assumptions and neural interpretability, raising transferability concerns; (2) Shuffling introduces new trust dependencies and communication overhead; (3) Most critically, all existing techniques ignore the structural coupling between phases, failing to address noise interdependence or inherent sequential defects of LDP.

Our work is the first to formally analyze this structural issue and avoid unnecessary utility loss through a novel framework, providing a principled solution beyond heuristic optimizations.

## 3 PRELIMINARIES

### 3.1 FEDERATED LEARNING

A basic FL system consists of one server and $N$ clients. Each interaction comprises two phases: uplink and downlink. In the uplink phase of the $t$-th interaction, client $i$ trains the model locally and then uploads the result $\mathbf{w}_i^t$ to the central server, which aggregates them as follows:

$$\mathbf{w}^t = \sum_{i=1}^{N} p_i \mathbf{w}_i^t, \tag{1}$$

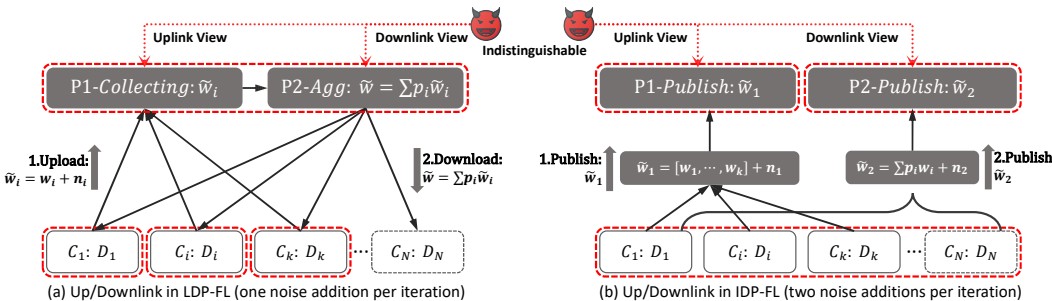

Figure 1: Interaction Modes of LDP & IDP. In LDP (a), datasets are distributed, the server selects $k$ clients to perform training with clients adding noise once. In IDP (b), all datasets are aggregated through a protocol, conducting two independent publishing to mimic uplink and downlink, where 2 kinds of noise are added separately, ensuring the privacy of both phases independently.

where $p_i = \frac{|D_i|}{\sum |D_i|}$ is the aggregation weight of client $i$, and $D_i$ denotes the dataset of the $i$-th client. In the downlink phase, the server sends $w^t$ back for the next round of training.

## 3.2 Differential Privacy in FL

Under the semi-honest setting, by measuring the maximum change rate of the query function, $(\epsilon, \delta)$-DP provides a strong criterion for privacy protection in FL.

**Definition 1** ($(\epsilon, \delta)$-DP(Dwork, 2006)). *A randomized mechanism $\mathcal{M} : \mathcal{X} \to \mathcal{R}$ with domain $\mathcal{X}$ and range $\mathcal{R}$ satisfies $(\epsilon, \delta)$-DP, if for all measurable sets $\mathcal{S} \subseteq \mathcal{R}$,*

$$\Pr[\mathcal{M}(D) \in \mathcal{S}] < e^\epsilon \Pr[\mathcal{M}(D') \in \mathcal{S}] + \delta,$$

*where $D, D' \in \mathcal{X}$ are any pair of adjacent datasets, which differ by only one record.*

**Definition 2** (Global Sensitivity (Dwork, 2006)). *: Given any dataset $D$ and query $f : D \to \mathcal{R}$, the global sensitivity $\Delta_f$ is defined as the maximum change in the query result,*

$$\Delta_f = \max_{D,D'} \|f(D) - f(D')\|_l,$$

*where $D, D'$ are any two adjacent datasets, differed by only one record, and $\|\cdot\|_l$ denotes the l-norm.*

Adding Gaussian noise $\mathbf{n} \sim \mathcal{N}(0, \sigma^2)$ to each dimension of all the queries can ensure $(\epsilon, \delta)$-DP, where the noise intensity $\sigma \geq \Delta \cdot s$, and $s = s(\epsilon, \delta, T)$ can be provided by the commonly used moment accountant (MA) (Wei et al., 2020).

## 3.3 Treat Model

In this paper, we inherit the general assumptions of LDP-FL: the server and clients are honest and will follow the protocol exactly. In addition, there exists an adversary who can infer clients' privacy by eavesdropping on the intermediate parameters $\{\mathbf{w}_i^t\}_{i=1}^N$ and $\mathbf{w}^t$ during FL procedure.

## 4 Revisiting Up & Downlink of DP-FL

In this section, we revisit the two phases of LDP and propose a new interaction mode, Ideal Differential Privacy (IDP). By proving that IDP accumulates less noise than LDP under the same privacy guarantees (Thm.1), we quantitatively present the inherent flaw of LDP, termed noise redundancy. In later sections, we present an instantiation framework (Alg.1) to demonstrate the feasibility of IDP.

### 4.1 Ideal vs Traditional Interaction Modes

In LDP, as shown in Fig.1(a), clients add noise $\mathbf{n}_i$ to local models $\mathbf{w}_i$ before uploading them:

$$\begin{cases} \tilde{\mathbf{w}}_i = \mathbf{w}_i + \mathbf{n}_i^{local} \\ \mathbf{n}_i^{local} \sim \mathcal{N}(0, \sigma_{local,i}^2). \end{cases}$$

After aggregation, the final model consists of two parts (model component and noise component):

$$\tilde{\mathbf{w}} = \sum_{i=1}^{N} p_i \tilde{\mathbf{w}}_i = \sum_{i=1}^{N} p_i \mathbf{w}_i + \sum_{i=1}^{N} p_i \mathbf{n}_i = \mathbf{w} + \mathbf{n}_{server},$$

where $\mathbf{n}_{server}$ also follows a Gaussian distribution. It is worth noting that in LDP, the noise in downlink is entirely determined by the noise injection in uplink: $\mathbf{n}_{server} = \sum_{i=1}^{N} p_i \mathbf{n}_i^{local}$. However, this coupling may lead to an inherent defect: when the noise accumulation in uplink phase exceeds the protection requirement of downlink phase, $\sigma_{server} > \sigma_{server}^{required}$, noise redundancy occurs.

Our intuition is straightforward: if there exists an interaction mode that decouples the inherently sequential two phases in DP-FL and provides independent privacy protection for each, then the potential mismatch can be avoided. Accordingly, we propose a new interaction mode, IDP (Def.3). As illustrated in Fig.1(b), we assume the existence of an ideal mechanism that can regard the server and all clients as a collective entity to perform a two-phase independent model publishing by adding noise twice. Specifically, we can independently adjust the noises $\mathbf{n}_1, \mathbf{n}_2$ for both phases and make it indistinguishable for an external adversary to differentiate between IDP and LDP based on the collected information.

**Definition 3** (Interaction Modes of LDP & IDP ). *DP-FL interaction mode is defined as a quintuple:*

$$\mathcal{I} \triangleq \{\mathcal{P}, \mathcal{D}, \theta, \mathcal{O}, \mathcal{M}\},$$

*where $\mathcal{P} = \{\{C_1, \cdots, C_k\}, S\}$ represents the set of participants in one iteration, including $k$ selected clients and a server; $\mathcal{D} = \{D_1, \cdots, D_k\}$ denotes their datasets; $\theta \in \{uplink, downlink\}$ serves as the indicator to distinguish between the uplink and downlink phases; $\mathcal{O} : D_i \to \mathbf{w}_i$ is the training optimizer; $\mathcal{M} = \{\mathcal{M}_{LDP}^{\theta}, \mathcal{M}_{IDP}^{\theta}\}$ represents the DP noise injectors for different modes.*

*The interaction mode in LDP can be formally defined as the following two indivisible sequential processes. In contrast, the IDP interaction mode can be formally defined by utilizing a trusted third-party server and then proceeding through independent uplink and downlink phases:*

| Federated Interaction Mode of LDP-FL | Federated Interaction Mode of IDP-FL |
|---|---|
| **if** $\theta = uplink$ **then** $\quad$ **P1-Upload:** $\forall i \in [k]$ $\quad \tilde{\mathbf{w}}_i^{LDP} = \mathcal{M}_{LDP}^{uplink}(\mathcal{O}(D_i)) \to S$ **if** $\theta = downlink$ **then** $\quad$ **P2-Download:** $\forall i \in [k]$ $\quad \tilde{\mathbf{w}}^{LDP} = \sum_{j \in [k]} \frac{|D_j|}{|D|} \tilde{\mathbf{w}}_j^{LDP} \to C_i$ | **Joint:** $C_1, \cdots, C_k, S \to [C_1, \cdots, C_k, S] = S'$ **Joint:** $D_1, \cdots, D_k \to [D_1, \cdots, D_k] = D$ **if** $\theta = uplink$ **then** $\quad$ **P1-Publish:** $\tilde{\mathbf{w}}_{1,i}^{IDP} = \mathcal{M}_{IDP}^{uplink}(\mathcal{O}(D_i)), \forall i \in [k]$ **if** $\theta = downlink$ **then** $\quad$ **P2-Publish:** $\tilde{\mathbf{w}}_2^{IDP} = \mathcal{M}_{IDP}^{downlink}(\sum_{i \in [k]} \frac{|D_i|}{|D|} \mathcal{O}(D_i))$ **Such that:** $\tilde{\mathbf{w}}_2^{IDP} = \sum_{i \in [k]} \tilde{\mathbf{w}}_{1,i}^{IDP}$ |

Intuitively, in IDP, the independent addition of noise in the two phases is more likely to match their respective protection requirements than in LDP.

## 4.2 Noise Redundancy in LDP-FL

To quantitatively reveal the inherent noise redundancy in LDP, we compare the noise magnitudes in uplink and downlink phases of LDP and IDP. We first present several lemmas, the proofs of which are provided in the appendix, and the final comparison results are presented in Thm.1.

Let $D_i$ be the dataset of $i$, $D_{min}$ and $D$ denote the smallest and total dataset in the current sampling round, respectively, $C$ is the clipping threshold such that the clipped gradient $\mathbf{g}$ satisfies $||clip(\mathbf{g}, C)||_2 = ||\mathbf{g}||_2 \cdot min\{1, C/||\mathbf{g}||_2\} \le C$.

**Lemma 1** (sensitivity of LDP-FL & IDP-FL). *In the traditional LDP-FL interaction mode, the sensitivity corresponding to the local client $i$ is:*

$$\Delta_i^{local} = \frac{2C}{|D_i|}.$$

*In IDP-FL interaction mode, the sensitivities corresponding to the uplink and downlink phases are:*

$$\Delta_{up} = \frac{2C}{|D_{min}|}, \quad \Delta_{down} = \frac{2C}{|D|}.$$

**Lemma 2** (combined properties (Wei et al., 2020)). *For $T$ different queries $f_t$ with sensitivity $\Delta_t$, to ensure satisfying $(\epsilon,\delta)$-DP, the intensity of the noise $\sigma_t$ from the Gaussian mechanism should satisfy:*

$$\sigma_t = \frac{\Delta_t \sqrt{2T \ln(1/\delta)}}{\epsilon}. \tag{2}$$

In IDP, performing federated aggregation $T$ times is equivalent to alternately conducting $T$ uplink and downlink publishing, resulting in a total of $2T$ queries. Therefore, we have:

$$\sigma_t = \frac{\Delta_t \sqrt{4T \ln(1/\delta)}}{\epsilon}, \ \Delta_t = \begin{cases} \Delta_{up}, & 2 \nmid t \\ \Delta_{down}, & 2 \mid t \end{cases}, \ t \in [1, 2T]$$

In DP-FL, fairness is typically ensured by having all clients agree on the same privacy parameters. However, slight deviations are sometimes allowed. Our approach is to fix a commonly used $\delta$ and reflect the difference in $\epsilon$. By Lemma.2, for any set of privacy parameters $(\epsilon_1, \delta_1)$, it can be transformed into equivalent parameters $(\epsilon_1 \sqrt{\frac{\ln(1/\delta)}{\ln(1/\delta_1)}}, \delta)$ while keeping the noise intensity unchanged. Furthermore, Lemma.3 presents the equivalent noise intensity in downlink phase of LDP.

**Lemma 3** (equivalent noise intensity of server in LDP). *In LDP, suppose that $k$ clients are sampled and each client $i$ satisfies $(\epsilon_i, \delta)$-DP with sensitivity $\Delta_i^{local} = \frac{2C}{|D_i|}$. Without loss of generality, assume that $(1 - \kappa)\epsilon = \epsilon_1 \leq \cdots \leq \epsilon_k = \epsilon$, $\kappa \ll 1$. Then LDP satisfies $(\epsilon, \delta)$-DP, and the equivalent noise intensity in downlink phase satisfies:*

$$\frac{8C^2 qTk \ln(1/\delta)}{\epsilon^2 |\mathcal{D}|^2} \leq \sigma_{server}^2 \leq \frac{8C^2 qTk \ln(1/\delta)}{(1 - \kappa)^2 \epsilon^2 |\mathcal{D}|^2},$$

*where $\kappa$ measures the variance of $\{\epsilon_i\}_{i \in [k]}$. For the sake of fairness, $\kappa$ is set to a very small value.*

With this series of preparations, we can compare the noise required by LDP and IDP. As a result, the following theorem quantitatively demonstrates that LDP suffers noise redundancy.

**Theorem 1** (noise redundancy). *Suppose a FL system with $N$ clients undergoes $T$ iterations. In each iteration, clients are sampled at a rate $q > \sqrt{2/N}$. To ensure $(\epsilon, \delta)$-DP, in IDP, clients add Gaussian noise with $\sigma_{up} = \frac{2C\sqrt{4T \ln(1/\delta)}}{\epsilon |\mathcal{D}_{min}|}$ in uplink, while $\sigma_{down} = \frac{2C\sqrt{4T \ln(1/\delta)}}{\epsilon |\mathcal{D}|}$ in downlink. Moreover, compared to LDP, IDP requires larger noise intensity in uplink, while smaller in downlink.*

*Proof.* In IDP mode, we abstract the two-stage process into two query functions, $f_{up}$ and $f_{down}$, with sensitivities $\Delta_{up}$ and $\Delta_{down}$ respectively. From Lemma.1, we have:

$$\Delta_{up} = \frac{2C}{|D_{\min}|}, \quad \Delta_{down} = \frac{2C}{|D|}.$$

According to Lemma.2, to ensure $(\epsilon, \delta)$-DP over a total $2T$ queries, the noise intensities corresponding to the two phases should be satisfied as follows:

$$\sigma_{up} = \frac{2C\sqrt{4T \ln(1/\delta)}}{\epsilon |D_{\min}|}, \ \sigma_{down} = \frac{2C\sqrt{4T \ln(1/\delta)}}{\epsilon |D|}. \tag{3}$$

Comparing the equivalent intensities in two phases under LDP mode, combining Lemma.3, we have:

$$\frac{\sigma_{server}^2}{\sigma_{down}^2} \geq \frac{8C^2 qTk \ln(1/\delta)}{\epsilon^2 |\mathcal{D}|^2} \cdot \frac{\epsilon^2 |\mathcal{D}|^2}{16C^2 T \ln(1/\delta)} = \frac{q^2 N}{2} > 1.$$

As for the noise intensity $\sigma_{local,i}^2$ added by client $i$ under LDP mode, from Lemma 2, we have:

$$\frac{\sigma_{local,i}^2}{\sigma_{up}^2} = \frac{8C^2 qT \ln(1/\delta)}{\epsilon_i^2 |D_i|^2} \cdot \frac{\epsilon^2 |D_{min}|^2}{16C^2 T \ln(1/\delta)} \leq \frac{q}{2(1 - \kappa)^2} < 1.$$

$\square$

According to Thm.1, in LDP, the noise in downlink exceeds the noise requirement in an ideal scenario. If there exists a framework that can provide independent privacy protection for both phases to simulate IDP, it would avoid noise redundancy caused by the sequential coupling of the two phases in LDP.

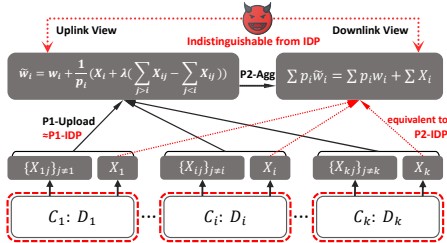

Figure 2: NADP simulates IDP. Each client generates two sets of independent random variables to produce counteracting and residual noises. The NADP framework is indistinguishable from IDP interaction mode.

## 5 NOISE-ANNIHILATION DP-FL

In this section, we present Noise Annihilation Differentially Private Federated Learning (NADP-FL), to instantiate IDP. Conceptually, we are inspired by theoretical physics: during annihilation, the mass of particles disappears and is converted into energy, with the total energy remaining unchanged. Similarly, we aim to reduce the accumulation of noise in the model by allowing a portion of the noise to "annihilate" each other while maintaining the same level of protection, which can make the noise in uplink and downlink phases mutually independent, thereby avoiding the issue of noise redundancy.

---

**Algorithm 1** NADP-FL

---

**Parties:** Clients $1, \ldots, N$ with datasets $D_i$, Server.
**Public Parameters:** model vector length $l$, input domain $\mathcal{X}^l$, standard Gaussian sequence sampler: $GS(seed) \mapsto \mathcal{X}^l \sim \mathcal{N}(0,1)$, public clipping threshold $C_t$, privacy parameters $(\epsilon, \delta)$, sampling rate $q$.
**Input:** $D_i$ (by each client $i$).
**Output:** $\mathbf{w}^T \in \mathcal{X}^l$ (by the server).

1: **Preparation Phase:**
2: Initialize: $t = 0$ and $\mathbf{w}^0$.
3: Each client $i$ generates secret key $s_i$, and uses Diffie-Hellman protocol to establish a shared key $s_{ij}$ with each client $j \neq i$, which serve as seeds.
4: **while** $t < T$ **do**
5:     Sample clients with probability $q$, obtaining $k$ clients. Sampled client set is denoted as $\mathcal{K}_t$.
6:     **Local Training Phase:**
7:     Broadcast $\mathbf{w}^t$ to all clients.
8:     **for** each client $i \in \mathcal{K}_t$ **do**
9:         Perform local training and clip gradients using $C_t$, obtain the local model $\mathbf{w}_i^{t+1}$.
10:         **Noise Adding Phase:**
11:         Allocate noise intensities with Alg.2, yielding $\{\sigma_i^{t+1}, \sigma_{ij}^{t+1}\}_{i \neq j \in [k]}$.
12:         Update secret values:
        $s_{ij}^{t+1} \leftarrow (s_{ij}||t),\ s_i^{t+1} \leftarrow (s_i||t),\ \forall j \in \mathcal{K}_t/\{i\}$.
13:         Sample noise set:
        $\mathbf{n}_{ij}^{t+1} \leftarrow \sigma_{ij}^{t+1} \cdot GS(s_{ij}^{t+1}),\ \forall j \in \mathcal{K}_t/\{i\}$,
        $\mathbf{n}_i^{t+1} \leftarrow \sigma_i^{t+1} \cdot GS(s_i^{t+1})$.
14:         Set noise compensation factor $\lambda$.
15:         Add noise:
        $p_i^{t+1} \leftarrow \frac{|D_i|}{\sum_{k \in \mathcal{K}_t} |D_k|}$,
        $\tilde{\mathbf{w}}_i^{t+1} \leftarrow \mathbf{w}_i^{t+1} + \frac{1}{p_i}(\mathbf{n}_i^{t+1} + \lambda(\sum_{j>i} \mathbf{n}_{ij}^{t+1} - \sum_{j<i} \mathbf{n}_{ij}^{t+1}))$.
16:         Upload model parameters to the server.
17:     **Model Aggregating Phase:**
18:     $\mathbf{w}^{t+1} \leftarrow \sum_{k \in \mathcal{K}_t} p_k^{t+1} \tilde{\mathbf{w}}_k^{t+1}$.
19:     $t \leftarrow t + 1$.
20: **return** $\mathbf{w}^T$

---

To implement this, we artificially decompose the noise added on client $i$ into two parts: $\mathbf{n}_i^{local} = \mathbf{n}_i^{residual} + \mathbf{n}_i^{counteract}$. $\mathbf{n}_i^{residual}$ is retained in the aggregated model to provide protection for downlink phase, while $\mathbf{n}_i^{counteract}$ is negotiated with the remaining $k-1$ clients, ensuring mutual cancellation during aggregation, thereby solely protecting the privacy in uplink phase, i.e., $\sum p_i \mathbf{n}_i^{counteract} = 0$.

Specifically, on the one hand, any two clients $i,j$ employ the Diffie-Hellman protocol (Diffie & Hellman, 2022) to negotiate shared secret keys $s_{ij} = s_{ji}$, which can be used as seeds to sample paired noise sequences $\mathbf{n}_{ij} = \mathbf{n}_{ji}$, Further, client $i$ scales each of $\{\mathbf{n}_{ij}\}_{j\neq i}$ to serve as local counteracting noise: $\mathbf{n}_i^{\text{counteract}} = \frac{1}{p_i}\sum_{j>i}\mathbf{n}_{ij} - \sum_{j<i}\mathbf{n}_{ij}$. The aggregation cancellation is ensured by (1) and the following identity:

$$\sum_{i=1}^{k}\Big(\sum_{j>i\in[k]}\mathbf{n}_{ij} - \sum_{j<i\in[k]}\mathbf{n}_{ij}\Big) = \mathbf{0}. \quad (4)$$

On the other hand, each client $i$ generates a local seed $s_i$ to sample the local residual noise: $\mathbf{n}_i^{residual} = \frac{1}{p_i}\mathbf{n}_i$. Moreover, the variance of the noises $n_i$ and $n_{ij}$ needs to be finely allocated to match the distribution of dataset sizes (see Section.6). In addition, we deploy a noise compensation coefficient $\lambda$ on local counteracting noise to enhance privacy at client level without further utility loss. Our framework is presented in Alg.1, which consists of four steps:

---

**Algorithm 2** Noise Intensity Allocation

**Input:** $\{|D_i|, p_i\}_{i\in[k]}, \sigma_{down}^2$.
1: Sort $\{|D_i|\}_{i\in[k]}$ in ascending order.
2: Calculate $\{\beta_i = |D_i|^2/|D_{\min}|^2 - p_i\}_{i\in[k]}$.
3: $\theta_1 \leftarrow \frac{1}{k-1}\beta_1$.
4: **for** $n \in [2, k-2]$ **do**
5: $\quad \theta_n \leftarrow \frac{1}{k-n}(\beta_n - \sum_{j=1}^{n-1}\theta_j)$.
6: $\theta_{k-1} \leftarrow \beta_k - \sum_{j=1}^{k-2}\theta_j$.
7: **for** $i \neq j \in [k]$ **do**
8: $\quad \sigma_i^2 \leftarrow p_i\sigma_{down}^2$.
9: $\quad \sigma_{ij}^2 \leftarrow \theta_i\sigma_{down}^2$, $\forall i < j$.
10: $\quad \sigma_{ij}^2 \leftarrow \theta_j\sigma_{down}^2$, $\forall i > j$.
11: **return** $\boldsymbol{\Sigma}^k = \{\sigma_i^2, \sigma_{ij}^2\}_{i\neq j\in[k]}$.

---

- **Preparation Phase:** Each client $i$ pairs up with others to negotiate common secrets $\{s_{ij}\}_{j\neq i}$, which serves as the seeds for generating synchronized counteracting noise. Additionally, each client generates another secret $s_i$ to serve as the seed for generating residual noise. The server then broadcasts the initial model $\tilde{\mathbf{w}}^0$.

- **Local Training Phase:** Sampled clients perform local training and apply gradient clipping.

- **Noise Adding Phase:** By performing noise intensity allocation (Alg.2, see Section.6), each client obtains the corresponding noise intensities for both counteracting and residual noise. Further, the clients link the iteration indicator $t$ to the secret values as updated seeds, from which $k - 1$ counteracting noises and one residual noise are sampled. Then, counteracting noises are amplified using the noise compensation coefficient $\lambda$ to enhance protection. Finally, all the noises are added, and the model is uploaded to the server.

- **Model Aggregating Phase:** The server weights the models and aggregates them.

As shown in Fig.2, in NADP, client $i$ generates $k - 1$ counteracting random variables $\{X_{ij}\}_{j\neq i}$ and generates a residual random variable $X_i$, where $\{X_{ij}\}_{j\neq i}$ and $X_i$ are independent of each other, which sample Gaussian sequences with different variances separately. The independence of the noise allows for separate privacy protection in the uplink and downlink phases. Specifically, let $Z_i^{up}, Z^{down}$ be the total noise variable added by client $i$ in uplink phase and the aggregated noise variable in downlink phase, respectively:

$$Z_i^{up} = \frac{1}{p_i}\Big(X_i + \lambda\big(\sum_{j>i}X_{ij} - \sum_{j<i}X_{ij}\big)\Big) \sim \mathcal{N}(0, \sigma_1^2),$$

$$Z^{down} = \sum_{i=1}^{k}p_i Z_i^{up} = \sum_{i=1}^{k}X_i \sim \mathcal{N}(0, \sigma_2^2).$$

Here, $\sigma_1^2 = Var(Z_i^{up}), \sigma_2^2 = Var(Z^{down})$, which can be independently controlled by adjusting $Var(X_i), \{Var(X_{ij})\}_{i\neq j}$ separately. This indicates that NADP simulates IDP interaction mode, thereby avoiding the issue of noise redundancy. It is worth noting that we introduce a noise compensation coefficient $\lambda$, which allows us to independently enhance the protection in the uplink phase. The additional noise introduced by this enhancement will also cancel out during aggregation (as shown in (4)), and thus will not incur further utility loss.

## 6 NOISE INTENSITY ALLOCATION

In this section, we explore the intensities of these noises to ensure $(\epsilon, \delta)$-DP. By setting $\lambda = 1$, we consider the standard intensity allocation required to defend against bystanders.

We assume an eavesdropper comprises both bystanders and the server, which can obtain both the gradients uploaded by all clients and the aggregated gradients distributed by the server.

In each iteration of NADP, we assume $k$ clients are sampled. Let the random variable $Z_i = \frac{1}{p_i}(\sum_{j>i} X_{ij} - \sum_{j<i} X_{ij} + X_i)$ represent the noise added by client $i$ (without $\lambda$). To ensure $(\epsilon, \delta)$-DP, the following conditions must be satisfied:

$$\begin{cases} p_i Z_i = \sum_{j>i} X_{ij} - \sum_{j<i} X_{ij} + X_i, \ i = 1, 2, \cdots, k \\ Z_i \sim \mathcal{N}(0, \sigma_{up}^2) \ , \ \sum_{i=1}^{k} X_i \sim \mathcal{N}(0, \sigma_{down}^2) \\ X_{ij} = X_{ji} \ , \ p_i = |D_i|/|D| \ , \end{cases}$$

where $\sigma_{up}$ and $\sigma_{down}$ represent the noise requirements for uplink and downlink phases in IDP, respectively (defined in (3)). Examining the variances of each term and employing a variable substitution: $Var(X_{ij}) = x_{ij} \cdot \frac{16 C^2 T \ln(1/\delta)}{\epsilon^2 |D|^2} = x_{ij} \cdot \sigma_{down}^2$ and $Var(X_i) = x_i \cdot \sigma_{down}^2$, we can obtain its simplified form:

$$\begin{cases} \sum_{j=1, j\neq i}^{k} x_{ij} + x_i = \frac{|D_i|^2}{|D_{min}|^2} \geq 1 \ , \ i = 1, 2, \cdots, k \\ \sum_{i=1}^{k} x_i = 1 \ , \ x_{ij} = x_{ji} \geq 0 \ , \ x_i \geq 0. \end{cases} \tag{5}$$

The noise intensity distribution induced by the solution of (5) provides a reasonable allocation for NADP to satisfy $(\epsilon, \delta)$-DP, from the perspective of bystanders. However, under certain extreme cases, finding a non-negative solution becomes challenging. To address this, we propose a method based on mathematical induction to obtain an approximate solution without introducing excessive noise.

**Theorem 2** (reasonableness of intensity allocation). *There exists an approximate solution $\theta^k = \{x_i, x_{ij}\}_{i\neq j\in[k]}$, where **only one** variable may violate the non-negativity condition in (5). Further, from this solution, a reasonable noise intensity allocation $\Sigma^k = \{\sigma_i^2, \sigma_{ij}^2\}_{i\neq j\in[k]}$ can be derived.*

We formally present Thm.2 in Alg.2, which provides an effective method for calculating a reasonable allocation. The proof is available in Appendix C. For Alg.2, on one hand, we set the only possible negative value $\beta_{k-1}$ equal to $\beta_k \geq 0$, which effectively increases the actual noise intensity for a single client but still ensures privacy. On the other hand, under this configuration, all allocations are made in pairs, allowing the counteracting noise to cancel out during aggregation and thereby preventing the introduction of excessive noise.

## 7 SIMULATION RESULTS

To most intuitively demonstrate the effectiveness of NADP-FL, we deploy the classic CNN on three representative CV datasets, MNIST (Deng, 2012), Fashion-MNIST (Xiao et al., 2017), FEM-NIST (Caldas et al., 2018) for image classification tasks, and deploy the GRU-RNN (Cho et al., 2014) on IMDb (Maas et al., 2011) for NLP task. We examine the utility superiority of NADP under different scales and privacy parameters, and also evaluate its performance in terms of scalability, reliability and privacy (presented in Appendix F, due to space constraints). In terms of the FL setup, we consider three scales of client numbers: large($N = 100$), medium($N = 50$), and small($N = 10$ or 25). Additionally, we set $\frac{|D_{max}|}{|D_{min}|} \leq 2$ (prevent excessive interference caused by large differences in data distribution) and ensure that under each scale, we randomly sample the same distribution of client dataset sizes. We set the total number of rounds $T = 200$, $\delta = 10^{-5}$, and $C = 10$, with one local training iteration per round. For the optimizer, we use SGD with a learning rate of 0.1 and a decay rate of 0.995. For all experiments, our primary comparison target is LDP with commonly used MA composition mechanism.

As shown in Table.1, our framework NADP outperforms LDP across different $\epsilon$. Under stronger privacy protection (smaller $\epsilon$), LDP-FL fails as the level of protection increases, however, NADP still

Table 1: Acc(%) of Different Frameworks under Different Scales with $q = 0.8$

| | $N$ | No-DP | $\epsilon = 3$ | | $\epsilon = 2$ | | $\epsilon = 1$ | | $\epsilon = 0.5$ | |
|---|---|---|---|---|---|---|---|---|---|---|
| | | | LDP | NADP | LDP | NADP | LDP | NADP | LDP | NADP |
| MNIST | 10 | 92.94 | 81.76 | 82.20 (0.44↑) | 80.27 | 82.05 (1.78↑) | 77.23 | 81.65 (4.42↑) | 67.44 | 78.41 (**10.97**↑) |
| | 50 | 93.30 | 83.94 | 85.55 (1.61↑) | 81.52 | 85.58 (4.06↑) | 65.47 | 85.19 (19.72↑) | 29.97 | 80.39 (**50.42**↑) |
| | 100 | 92.91 | 85.09 | 87.37 (2.28↑) | 76.69 | 87.33 (10.64↑) | 46.97 | 86.84 (39.87↑) | 10.44 | 80.58 (**70.14**↑) |
| FMNIST | 10 | 80.46 | 80.02 | 80.84 (0.82↑) | 79.08 | 80.48 (1.40↑) | 76.34 | 79.29 (2.95↑) | 70.76 | 76.88 (**6.12**↑) |
| | 50 | 82.44 | 78.90 | 81.61 (2.71↑) | 77.12 | 81.36 (4.24↑) | 71.16 | 80.17 (9.01↑) | 60.21 | 77.19 (**16.98**↑) |
| | 100 | 83.23 | 77.97 | 82.33 (4.36↑) | 74.88 | 82.02 (7.14↑) | 67.41 | 81.08 (13.67↑) | 52.81 | 77.71 (**24.90**↑) |
| EMNIST | 10 | 88.68 | 78.57 | 78.56 (0.01↓) | 78.01 | 78.65 (0.64↑) | 75.57 | 77.78 (2.21↑) | 67.30 | 74.94 (**7.64**↑) |
| | 50 | 88.65 | 76.35 | 78.68 (2.33↑) | 74.18 | 78.49 (4.31↑) | 63.49 | 77.47 (13.98↑) | 44.45 | 74.32 (**29.87**↑) |
| | 100 | 88.61 | 75.51 | 78.67 (3.16↑) | 70.68 | 78.63 (7.95↑) | 54.30 | 77.98 (23.68↑) | 31.05 | 75.07 (**44.02**↑) |
| IMDb | 10 | 89.71 | 85.32 | 87.24 (1.92↑) | 82.93 | 85.29 (2.36↑) | 79.46 | 83.79 (4.33↑) | 59.32 | 78.22 (**18.9**↑) |
| | 50 | 88.56 | 84.97 | 86.68 (1.71↑) | 79.86 | 84.80 (4.94↑) | 68.37 | 84.61 (16.24↑) | 50.68 | 80.29 (**29.61**↑) |
| | 100 | 90.87 | 84.25 | 87.09 (2.84↑) | 78.19 | 84.14 (5.95↑) | 61.89 | 83.98 (22.09↑) | 49.19 | 79.63 (**30.44**↑) |

maintains a high utility. It is worth noting that, as $N$ increases, NADP experiences almost no performance degradation, whereas the accuracy of LDP drops rapidly. This highlights the greater potential of our framework in large-scale scenarios. Additional experiments are presented in Appendix F.

## 8 DISCUSSION

**Overheads Discussion:** Without considering the computational overhead of local training, we compare the computation, communication, and storage overheads of LDP, NADP, and secure aggregation (SA) (Bonawitz et al., 2017) in each round, where $m$ denotes the size of model, $n$ represents the number of clients, $n \ll m$. The results are presented in Table 2.

Table 2: Complexity among Different Frameworks.

| Cost\Framework | LDP-FL | NADP-FL | SA-FL |
|---|---|---|---|
| Client Computation | $O(m)$ | $O(n\log(n)+nm)$ | $O(n^2+nm)$ |
| Client Communication | $O(m)$ | $O(m)$ | $O(m + n)$ |
| Client Storage | $O(m)$ | $O(m + n)$ | $O(m + n)$ |
| Server Computation | $O(nm)$ | $O(nm)$ | $O(n^2m)$ |
| Server Communication | $O(nm)$ | $O(nm)$ | $O(n^2+nm)$ |
| Server Storage | $O(nm)$ | $O(nm)$ | $O(n^2+nm)$ |

Compared to LDP, as key negotiation and other preliminary operations are conducted in advance, the primary additional cost of NADP lies in local computation, which is reflected in the generation of $n$ noise sequences and the execution of Alg.2. On the one hand, Alg.2 consists of a sorting algorithm ($O(n\log(n))$) and $n$ iterations of computation ($O(n)$), and since all computational objects are scalars ($\theta_i$), additional computation is minimal; on the other hand, the sampling of $n$ noises is also completed locally, which prevents a significant decrease in the computational efficiency. Moreover, in contrast to SA, NADP incurs lower overhead due to the absence of secret-sharing for each round.

**Advancing Toward Malicious Scenarios:** We follow the general assumption in DP-FL framework that all clients honestly execute the protocol. However, even without this assumption, under scenarios similar to LDP that do not consider malicious poisoning, NADP will not collapse. The discussion details are presented in Appendix A.

## 9 CONCLUSION

In this paper, we revisit the two-stage interaction mode of uplink and downlink in LDP-FL and propose a novel interaction perspective, IDP, to demonstrate the inherent flaw in LDP, called noise redundancy. On this basis, we instantiate IDP and propose NADP-FL, which decouples the noise intensity requirements of uplink and downlink phases using counteracting noise to mitigate the utility loss. Furthermore, we discuss the intensity allocation to counteract noise to ensure $(\epsilon, \delta)$-DP. Finally, we validate the superiority of our framework in terms of utility, scalability, robustness through extensive experiments. Both theoretical and experimental results demonstrate that our framework achieves a better utility-privacy trade-off than the traditional LDP-FL framework.

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

## A ADVANCING TOWARD MALICIOUS SCENARIOS

In malicious settings, under scenarios similar to LDP that do not consider malicious poisoning, NADP will not collapse. On one hand, the impact of malicious client $i$ on the noise generated by any benign client $j$ is limited: it can only control $n_{ij} \neq n_{ji}$, preventing the associated noise from being canceled out, but it will not affect the noise cancellation among the remaining benign clients. This is similar to the dropout situation, and as we also show in Figure 4 (Appendix F), the impact of incomplete noise cancellation on utility is limited. On the other hand, when malicious clients add

non-Gaussian noise, $Z_i^{up}$ indeed no longer follows a Gaussian distribution. However, introducing $\lambda$ and budgeting can still satisfy $(\epsilon, \delta)$-DP privacy for benign clients, and the noise redundancy introduced by malicious clients will further enhance privacy.

We next consider a typical scenario in a malicious setting. Specifically, we significantly enhance the adversary's capabilities in Section 6: the adversary can also collude and collect all the local information from a subset of participants to infer the privacy of the remaining benign clients. However, unlike the traditional collusion scenario in the malicious setting, the adversary cannot manipulate the eavesdropped clients, who still remain honest. We assert that appropriately selecting the noise compensation coefficient $\lambda$ can protect against such attacks.

Assuming an adversary $\mathcal{A}_2$ that corrupts the server and a proportion $\tau$ of the clients sampled in each iteration. Let $G = \{1, 2, \cdots, (1-\tau)k\}$ and $C = \{(1-\tau)k+1, (1-\tau)k+2, \cdots, k\}$ denotes the set of indices corresponding to honest and corrupted clients, respectively. The adversary $\mathcal{A}_2$ can generate two types of inference queries $f_{up}^{adv}, f_{down}^{adv}$ through uplink and downlink phases by subtracting the information already obtained:

$$f_{up}^{adv}(D_G) = f_1(D/D_C) = [g_1, \ldots, g_{(1-\tau)k}, 0, \ldots, 0],$$
$$f_{down}^{adv}(D_G) = f_2(D) - f_2(D_C) = \sum_{i \in D_G} p_i g_i$$
$$= \sum_{i \in D_G} \frac{|D_i|}{|D|} \cdot \frac{1}{|D_i|} \sum_{j=1}^{|D_i|} g_i^j = \frac{1}{|D|} \sum_{j=1}^{|D_G|} g^j.$$

Here, $D_C$ and $D_G$ denote the total datasets of the corrupted clients and the honest clients, respectively, and $g^j$ denotes the gradient computed from the $j$-th data in $D_G$. Analogous to the proof of Lemma 1, we assume that the difference in neighboring datasets exists in $D_1$, then we can get:

$$\Delta_{up}^{adv} = \max_{D_G, D_G'} ||f_{up}^{adv}(D_G) - f_{up}^{adv}(D_G')||_2$$
$$= \max_{D_G, D_G'} ||[g_1' - g_1, \cdots, g_{(1-\tau)k}' - g_{(1-\tau)k}, 0, \cdots, 0]||_2$$
$$= \max_{D_G, D_G'} ||g_1' - g_1||_2 \le \frac{2C}{|D_{min}|},$$
$$\Delta_{down}^{adv} = \max_{D_G, D_G'} ||f_{down}^{adv}(D_G) - f_{down}^{adv}(D_G')||_2$$
$$= \max_{D_G, D_G'} ||\frac{1}{|D|} \sum_{i=1}^{|D_G|} g^i - \frac{1}{|D|+1} (\sum_{i=1}^{|D_G|} g^i + g')||_2$$
$$\le \max_{D_G, D_G'} \frac{\sum_{i=1}^{|D_G|} ||g^i - g'||_2}{|D|(|D|+1)} \le \frac{2C|D_G|}{|D|^2} = \frac{2C}{|D|} \sum_{i \in G} p_i.$$

Therefore, according to Lemma 2, to ensure the system satisfies $(\epsilon, \delta)$-DP for inference queries $f_{up}^{adv}$ and $f_{down}^{adv}$, the noise intensities should satisfy:

$$\sigma_{adv,up}^2 = \frac{16C^2 T \ln(1/\delta)}{\epsilon^2 |D_{min}|^2} = \sigma_{up}^2,$$
$$\sigma_{adv,down}^2 = \frac{16C^2 T \ln(1/\delta)}{\epsilon^2 |D|^2} (\sum_{i \in G} p_i)^2 < \sigma_{down}^2.$$

In addition, for an honest client $i$, the effective noise intensities from the adversary's perspective (the remaining after canceling out the adversary's portion) are:

$$\sigma^2_{G,up,i} = \sigma^2_{up} - \sum_{j \in C} \sigma^2_{ij} < \sigma^2_{up} = \sigma^2_{adv,up},$$

$$\sigma^2_{G,down} = \sum_{i \in G} \sigma^2_i = \sum_{i \in G} p_i \sigma^2_{down}$$

$$= \frac{16C^2 T \ln(1/\delta)}{\epsilon^2 |D|^2} \sum_{i \in G} p_i > \sigma^2_{adv,down}.$$

Therefore, it can be observed that if collusion exists, the residual noise generated by honest clients satisfies the server-side security requirements from the internal adversaries, attributed to our intricate construction of the particular solution $\{x_i = p_i\}_{i \in [k]}$ in Thm.2.

Since the noise intensity is insufficient to meet client-side privacy requirements in collusion scenarios, we utilize $\lambda$ to enhance protection during the uplink phase.

For an honest client $i$, the actual effective noise it adds during the uplink phase is sampled by the random variable $Z'_i = \sum_{j>i,j \in G} X_{ij} - \sum_{j<i,j \in G} X_{ij} + X_i$. On this basis, we amplify the effective counteracting noise by $\lambda$ to meet the privacy requirements, i.e.,

$$Z_i = \lambda(\sum_{j>i} X_{ij} - \sum_{j<i} X_{ij}) + X_i.$$

To determine the range of $\lambda$, we examine the most extreme case of the intensity distribution. Adopting notations from Thm.2, we first examine the lower bound of $\alpha_2$:

$$\alpha_1 = \frac{|D|}{|D_{min}|} \leq \frac{|D_{min}| + (k-1)|D_{max}|}{|D_{min}|} = 1 + (k-1)\alpha_2. \tag{6}$$

Then, we examine the distribution of $\beta_i = \frac{|D_i|^2}{|D_{min}|^2} - \frac{|D_i|}{|D|}$ (defined in Alg.2). For $\beta_{min}$ and $\beta_{max}$, we have:

$$1 - \frac{1}{k} \leq \beta_{min} = 1 - \frac{|D_{min}|}{|D|} = 1 - \frac{1}{\alpha_1} < 1,$$

$$1 - \frac{1}{k} \leq \beta_{max} = \frac{|D_{max}|^2}{|D_{min}|^2} - \frac{|D_{max}|}{|D|} = \alpha_2^2 - \frac{\alpha_2}{\alpha_1}.$$

Where, $\alpha_2^2 - \frac{\alpha_2}{\alpha_1}$ is a quadratic function in terms of $\alpha_2$. Since $\alpha_2 \geq 1 > \frac{1}{2\alpha_1}$, $\beta_{max}$ is monotonically increasing respect to $\alpha_2$. Therefore, combining equation 6, we have:

$$\beta_{max} = \alpha_2^2 - \frac{\alpha_2}{\alpha_1} \leq \alpha_2^2 - \frac{1}{\frac{1}{\alpha_2} + (k-1)} < \alpha_2^2 - \frac{1}{k}.$$

Combining Alg.2, $1 - \frac{1}{k} \leq \beta_1 \leq \cdots \leq \beta_k < \alpha_2^2 - \frac{1}{k}$, we have:

$$\begin{cases} \theta_1 = \dfrac{1}{k-1}\beta_1, \\[2mm] \theta_n = \dfrac{1}{k-n}(\beta_n - \sum_{i=1}^{n-1} \theta_i), \ 2 \leq n \leq k-2, \\[2mm] \theta_{k-1} = \beta_k - \sum_{i=1}^{k-2} \theta_i. \end{cases}$$

Here, $[\theta_1, \theta_2, \cdots, \theta_{t-1}, 0, \theta_t, \cdots, \theta_t]$, $(\theta_i \leq \theta_{i+1})$ is the counteracting noise intensity allocation of client $t$ (see more details in Appendix E), and the allocation matrix is as follows:

$$\begin{bmatrix} 0 & \theta_1 & \theta_1 & \cdots & \theta_1 & \theta_1 \\ \theta_1 & 0 & \theta_2 & \cdots & \theta_2 & \theta_2 \\ \theta_1 & \theta_2 & 0 & \cdots & \theta_3 & \theta_3 \\ \vdots & \vdots & \vdots & \ddots & \vdots & \vdots \\ \theta_1 & \theta_2 & \theta_3 & \cdots & 0 & \theta_{k-1} \\ \theta_1 & \theta_2 & \theta_3 & \cdots & \theta_{k-1} & 0 \end{bmatrix} \begin{bmatrix} 1 \\ 1 \\ 1 \\ \vdots \\ 1 \\ 1 \end{bmatrix} = \begin{bmatrix} \beta_1 \\ \beta_2 \\ \beta_3 \\ \vdots \\ \beta_{k-1} \\ \beta_k \end{bmatrix}$$

To compute $\lambda(\tau)$, we consider the worst-case scenario, corresponding to the most extreme counter-acting noise intensity allocation $[\theta_1, \theta_2, \cdots, \theta_{k-1}, 0]$ (i.e. client $k$). Under this allocation, to ensure privacy protection, $\lambda$ needs to satisfy the following inequality:

$$\lambda^2(\theta_1 + \theta_2 + \cdots + \theta_{(1-\tau)k-1}) \geq \beta_k. \tag{7}$$

Since $\theta_i \leq \theta_{i+1}$, in order for equation 7 to hold, we set $\lambda$ to satisfy:

$$\lambda^2[(1-\tau)k - 1] \cdot \theta_1 = \lambda^2[(1-\tau)k - 1] \cdot \frac{\beta_1}{k-1} \geq \beta_k.$$

For $\beta_1 = \beta_{min} \geq 1 - \frac{1}{k}$ and $\beta_k = \beta_{max} < \alpha_2^2 - \frac{1}{k}$, we obtain:

$$\lambda \geq \sqrt{\frac{k \cdot \alpha_2^2 - 1}{(1-\tau)k - 1}}. \tag{8}$$

This indicates that $\lambda$ is positively correlated with $1/(1-\tau)$ in a sub-linear manner, ensuring that $\lambda$ does not become excessively large. In general, to ensure fairness, $\alpha_2 = \frac{|D_{max}|}{|D_{min}|}$ cannot be set too large. We suggest assuming that $\alpha_2 \leq \sqrt{2}$, which allows for up to 41% database size imbalance (for cases where $\alpha_2 > \sqrt{2}$, we can sample the data in each iteration of these clients that exceed the threshold, satisfying the fairness requirements of FLZhang et al. (2020)). Table 3 presents the magnitude of $\lambda$ under different settings.

Table 3: $\lambda$ under Different $\frac{|D_{max}|}{|D_{min}|}$, $K$ and $\tau$

| $\frac{|D_{max}|}{|D_{min}|}^2$ | K | $\lambda$ | | | | |
|---|---|---|---|---|---|---|
| | | $\tau = 0.1$ | $\tau = 0.2$ | $\tau = 0.3$ | $\tau = 0.4$ | $\tau = 0.5$ |
| 1.0 | 25 | 1.0565 | 1.1239 | 1.2060 | 1.3093 | 1.4446 |
| | 50 | 1.0553 | 1.1209 | 1.2005 | 1.2999 | 1.4289 |
| | 100 | 1.0547 | 1.1194 | 1.1978 | 1.2954 | 1.4214 |
| | $\infty$ | 1.0541 | 1.1180 | 1.1952 | 1.2910 | 1.4142 |
| 1.5 | 25 | 1.3029 | 1.3860 | 1.4873 | 1.6147 | 1.7815 |
| | 50 | 1.2968 | 1.3775 | 1.4753 | 1.5974 | 1.7559 |
| | 100 | 1.2939 | 1.3733 | 1.4695 | 1.5892 | 1.7438 |
| | $\infty$ | 1.2910 | 1.3693 | 1.4639 | 1.5811 | 1.7321 |
| 2.0 | 25 | 1.5097 | 1.6059 | 1.7233 | 1.8708 | 2.0642 |
| | 50 | 1.5000 | 1.5933 | 1.7064 | 1.8476 | 2.0310 |
| | 100 | 1.4953 | 1.5871 | 1.6983 | 1.8365 | 2.0152 |
| | $\infty$ | 1.4907 | 1.5811 | 1.6903 | 1.8257 | 2.0000 |

# B   PROOF OF LEMMA 1

*Proof.* According to the definition of LDP, we consider the neighboring datasets $D_i$, $D_i'$ for the $i$-th client, where $D_i$ has one fewer sample than $D_i'$ ($|D_i| = m_i$), with the remaining samples being the same. Therefore, the output of one training process can be written in the following form:

$$g_i = \frac{1}{m_i} \sum_{j=1}^{m_i} g_i^j.$$

According to the sensitivity calculation formula:

$$\Delta_i^{local} = \max_{D_i, D_i'} ||g_i(D_i) - g_i(D_i')||$$

$$= \max_{D_i, D_i'} ||\frac{1}{m_i} \sum_{j=1}^{m_i} g_i^j - \frac{1}{m_i + 1} \sum_{j=1}^{m_i} (g_i^j + g')||$$

$$\leq \max_{D_i, D_i'} \frac{\sum_{j=1}^{m_i} ||g_i^j - g'||}{m_i(m_i + 1)} \leq \frac{2C}{|D_i|}.$$

In IDP-uplink process, let the publish function corresponding to clients be $f_1(D_t) = [g_1, g_2, \ldots, g_k]$, where $D_t$ is the union of the $k$ clients sampled in $t$-th iteration, denoted as $D_t = D_1 \cup D_2 \cup \cdots \cup D_k$. Without loss of generality, let $D'_t = D'_1 \cup D_2 \cup \cdots \cup D_k$. Then, we have:

$$
\begin{aligned}
\Delta_{up} &= \max_{D_t, D'_t} ||f_{up}(D_t) - f_{up}(D'_t)||_2 \\
&= \max_{D_t, D'_t} ||[g'_1 - g_1, g'_2 - g_2, \ldots, g'_k - g_k]||_2 \\
&= \max_{D_t, D'_t} ||g'_1 - g_1||_2 \leq \frac{2C}{|D_{min}|}.
\end{aligned}
$$

In IDP-downlink process, the union collects the gradient intermediate results from each client, $g_i = \frac{1}{m_i} \sum_{j=1}^{m_i} g_i^j$, and performs a weighted average, finally outputting the aggregated result $g = \sum_{j=1}^{k} p_j g_j$, where $p_j = \frac{m_j}{\sum m_i}$. Therefore, for the equivalent overall training dataset $D\left(|D| = \sum m_i\right)$, one round of the aggregation process can be written as:

$$
f_{down}(D) = \sum_{i=1}^{k} p_i g_i = \sum_{i=1}^{k} \frac{p_i}{m_i} \sum_{j=1}^{m_i} g_i^j = \frac{1}{\sum m_i} \sum_{i,j} g_i^j,
$$

then the sensitivity is:

$$
\begin{aligned}
\Delta_{down} &= \max_{D, D'} ||f_{down}(D) - f_{down}(D')|| \\
&= \max_{D, D'} ||\frac{1}{\sum m_i} \sum g_i^j - \frac{1}{\sum m_i + 1}(\sum g_i^j + g')|| \\
&\leq \max_{D, D'} \frac{\sum ||g_i^j - g'||}{\sum m_i(\sum m_i + 1)} \leq \frac{2C}{|\mathcal{D}|}.
\end{aligned}
$$

$\square$

## C PROOF OF LEMMA 2

*Proof.* We follow the proof framework from (Wei et al., 2020), focusing only on the differences. For more details, please refer to the proof of Thm.1 in (Wei et al., 2020).

Define the privacy loss random variable for the t-th query as $L_t$. Then:

$$
L_t = \ln \frac{\Pr[\mathcal{M}_t(D) = o]}{\Pr[\mathcal{M}_t(D') = o]},
$$

where $\mathcal{M}_t(D) = f_t(D) + \mathcal{N}(0, \sigma_t^2)$.

For the Gaussian mechanism, the probability density functions of the output $o$ for the neighboring datasets are:

$$
\Pr[\mathcal{M}_t(D) = o] = \frac{1}{\sqrt{2\pi}\sigma_t} \exp\left(-\frac{(o - f_t(D))^2}{2\sigma_t^2}\right),
$$

$$
\Pr[\mathcal{M}_t(D') = o] = \frac{1}{\sqrt{2\pi}\sigma_t} \exp\left(-\frac{(o - f_t(D'))^2}{2\sigma_t^2}\right).
$$

Substituting this into the definition of privacy loss yields:

$$
L_t = \ln \frac{\frac{1}{\sqrt{2\pi}\sigma_t} \exp\left(-\frac{(o - f_t(D))^2}{2\sigma_t^2}\right)}{\frac{1}{\sqrt{2\pi}\sigma_t} \exp\left(-\frac{(o - f_t(D'))^2}{2\sigma_t^2}\right)}.
$$

By simplifying the calculation, we obtain:

$$
L_t = \frac{(o - f_t(D'))^2 - (o - f_t(D))^2}{2\sigma_t^2}.
$$

Let us denote $\Delta_t = f_t(D) - f_t(D')$, which can be controlled to be a small quantity. Then we have:

$$L_t = \frac{(o - f_t(D))\Delta_t}{\sigma_t^2} + \frac{\Delta_t^2}{2\sigma_t^2}.$$

In Gaussian mechanism, the output $o$ can be expressed as:

$$o = f_t(D) + \mathcal{N}(0, \sigma_t^2).$$

Then, further simplifying, the privacy loss becomes:

$$L_t = \frac{\Delta_t}{\sigma_t} \cdot Z_t + \frac{\Delta_t^2}{2\sigma_t^2} \sim N(\frac{\Delta_t^2}{2\sigma_t^2}, \frac{\Delta_t^2}{\sigma_t^2}),$$

where $Z_t \sim \mathcal{N}(0, 1)$ denotes a random variable from the standard normal distribution.

Consider the moment generating function (MGF) of the Gaussian-distributed variable, we have:

$$M_{L_t}(\lambda) = \mathbb{E}[\exp(\lambda L_t)] = \exp\left(\frac{\lambda(\lambda+1)\Delta_t^2}{2\sigma_t^2}\right).$$

For the noise intensity, we naturally constrain it to be proportional to the sensitivity, that is: $\frac{\Delta_i^2}{\sigma_i^2} = \frac{\Delta_j^2}{\sigma_j^2} = \frac{\Delta^2}{\sigma^2}$. Suppose $T$ independent queries are performed, then the MGF for multiple rounds of queries is:

$$M_{L_{\text{total}}}(\lambda) = \prod_{t=1}^{T}[M_{L_t}(\lambda)] = \exp\left(\sum_{t=1}^{T}\frac{\lambda(\lambda+1)\Delta_t^2}{2\sigma_t^2}\right)$$
$$= \exp\left(\frac{T\lambda(\lambda+1)\Delta^2}{2\sigma^2}\right).$$

Using the tail bound by moments (Abadi et al., 2016), we have:

$$\delta \geq \min_{\lambda} \exp(\frac{\lambda(\lambda+1)T\Delta^2}{2\sigma^2} - \lambda\epsilon) \in (0, 1). \tag{9}$$

Minimizing the RHS with respect to $\lambda$, we obtain:

$$\lambda = \frac{\epsilon\sigma^2}{T\Delta^2} - \frac{1}{2}.$$

Substituting into (9), we obtain:

$$\delta \geq \exp(-\frac{\epsilon^2\sigma^2}{2T\Delta^2} - \frac{T\Delta^2}{8\sigma^2} + \frac{\epsilon}{2}).$$

Combining with (9), we have:

$$\frac{\lambda(\lambda+1)T\Delta^2}{2\sigma^2} - \lambda\epsilon \leq 0.$$

Thus, we have:

$$\delta \geq \exp(-\frac{\epsilon^2\sigma^2}{2T\Delta^2} + \frac{2\lambda+1}{4\lambda+1}\epsilon) \geq \exp(-\frac{\epsilon^2\sigma^2}{2T\Delta^2}).$$

That is:

$$\sigma_t = \frac{\Delta_t\sqrt{2T\ln(1/\delta)}}{\epsilon}.$$

$\square$

# D   PROOF OF LEMMA 3

*Proof.* From Lemma 2, in LDP-FL, we have:

$$\sigma_{local,i} = \frac{\Delta_i \sqrt{2qT\ln(1/\delta)}}{\epsilon_i} = \frac{2C\sqrt{2qT\ln(1/\delta)}}{\epsilon_i |D_i|}.$$

Then, the equivalent noise intensity for the central server is:

$$\sigma^2_{server} = \sum_{i=1}^{k} p_i^2 \sigma^2_{local,i} = \sum_{i=1}^{k} (\frac{|D_i|}{\sum_{i=1}^{k}|D_i|})^2 \sigma^2_{local,i}$$

$$= \sum_{i=1}^{k} \frac{8C^2 qT\ln(1/\delta)}{\epsilon_i^2 |D|^2} = \frac{8C^2 qT\ln(1/\delta)}{|D|^2} \sum_{i=1}^{k} \frac{1}{\epsilon_i^2}.$$

Thus, we obtain:

$$\frac{8C^2 qTk\ln(1/\delta)}{\epsilon^2 |D|^2} \le \sigma^2_{server} \le \frac{8C^2 qTk\ln(1/\delta)}{(1-\kappa)^2 \epsilon^2 |D|^2}.$$

$\square$

# E   PROOF OF THEOREM 2

*Proof.* We only need to find a particular solution to (5). Let $x_i = p_i$, the equation can be transformed into:

$$\begin{cases} \sum_{j=1,j\neq i}^{k} x_{ij} = \frac{|D_i|^2}{|D_{min}|^2} - \frac{|D_i|}{|D|} \\ x_{ij} = x_{ji} \ge 0 \end{cases}$$

Let $\alpha_1 = \frac{|D|}{|D_{min}|} \ge k$, then we have:

$$\frac{|D_i|^2}{|D_{min}|^2} - \frac{|D_i|}{|D|} = p_i^2 \alpha_1^2 - p_i.$$

This is a quadratic function in terms of $p_i$. Since $p_i \ge \frac{|D_{min}|}{|D|} = \frac{1}{\alpha_1} > \frac{1}{2\alpha_1^2}$, $p_i$ is positively correlated with $\frac{|D_i|^2}{|D_{min}|^2} - \frac{|D_i|}{|D|}$. Thus, we can obtain:

$$\frac{|D_i|^2}{|D_{min}|^2} - \frac{|D_i|}{|D|} \ge \frac{|D_{min}|^2}{|D_{min}|^2} - \frac{|D_{min}|}{|D|} \ge 1 - \frac{1}{k} > 0.$$

Let $\beta_i = \frac{|D_i|^2}{|D_{min}|^2} - \frac{|D_i|}{|D|}$, $x_{ij} = b_{ij}$, where $b_{ij} \ge 0$. The above equation can be simplified to:

$$\begin{cases} \sum_{j=1,j\neq i}^{k} b_{ij} = \beta_i \quad i = 1,2,3,\cdots,k \\ b_{ij} = b_{ji} \ge 0 \end{cases}$$

Transform the system of equations into the corresponding matrix representation. Due to the symmetry, without loss of generality, assume $0 < \beta_1 \le \beta_2 \le \cdots \le \beta_k$, we have:

$$\begin{bmatrix} 0 & b_{12} & b_{13} & b_{14} & \cdots & b_{1(k-1)} & b_{1k} \\ b_{21} & 0 & b_{23} & b_{24} & \cdots & b_{2(k-1)} & b_{2k} \\ b_{31} & b_{32} & 0 & b_{34} & \cdots & b_{3(k-1)} & b_{3k} \\ b_{41} & b_{42} & b_{43} & 0 & \cdots & b_{4(k-1)} & b_{4k} \\ \vdots & \vdots & \vdots & \vdots & \ddots & \vdots & \vdots \\ b_{(k-1)1} & b_{(k-1)2} & b_{(k-1)3} & b_{(k-1)4} & \cdots & 0 & b_{(k-1)k} \\ b_{k1} & b_{k2} & b_{k3} & b_{k4} & \cdots & b_{k(k-1)} & 0 \end{bmatrix} \begin{bmatrix} 1 \\ 1 \\ 1 \\ 1 \\ \vdots \\ 1 \\ 1 \end{bmatrix} = \begin{bmatrix} \beta_1 \\ \beta_2 \\ \beta_3 \\ \beta_4 \\ \vdots \\ \beta_{k-1} \\ \beta_k \end{bmatrix} \tag{10}$$

We discuss the solution $\theta^k$ using mathematical induction:

$(i)$ When $k = 3$, the system of equations has a solution:

$$
\begin{cases}
b_{12} = b_{21} = \dfrac{\beta_1 + \beta_2 - \beta_3}{2} \\[2mm]
b_{13} = b_{31} = \dfrac{\beta_1 - \beta_2 + \beta_3}{2} \\[2mm]
b_{23} = b_{32} = \dfrac{-\beta_1 + \beta_2 + \beta_3}{2}
\end{cases}
\tag{11}
$$

Since $\beta_i \le \beta_{i+1}$, $b_{13} = b_{31} \ge 0$ and $b_{23} = b_{32} \ge 0$, that is, (11) has at most two variables $b_{23} = b_{32}$ less than $0$.

$(ii)$ We assume the existence of solution $\theta^k$ such that at most only one variable is less than zero for $k < K$.

$(iii)$ For the case when $k = K$, we let $b_{1i} = \frac{1}{K-1}\beta_1$. Then, the system of (10) becomes:

$$
\begin{bmatrix}
0 & b_{23} & b_{24} & \cdots & b_{2(K-2)} & b_{2K} \\
b_{32} & 0 & b_{34} & \cdots & b_{3(K-1)} & b_{3K} \\
b_{42} & b_{43} & 0 & \cdots & b_{4(K-1)} & b_{4K} \\
\vdots & \vdots & \vdots & \ddots & \vdots & \vdots \\
b_{(K-1)2} & b_{(K-1)3} & b_{(K-1)4} & \cdots & 0 & b_{(K-1)K} \\
b_{K2} & b_{K3} & b_{K4} & \cdots & b_{K(K-1)} & 0
\end{bmatrix}
\begin{bmatrix}
1 \\ 1 \\ 1 \\ \vdots \\ 1 \\ 1
\end{bmatrix}
=
\begin{bmatrix}
\beta_2 - \frac{1}{K-1}\beta_1 \\
\beta_3 - \frac{1}{K-1}\beta_1 \\
\beta_4 - \frac{1}{K-1}\beta_1 \\
\vdots \\
\beta_{K-1} - \frac{1}{K-1}\beta_1 \\
\beta_K - \frac{1}{K-1}\beta_1
\end{bmatrix}
$$

Since $0 \le \beta_2 - \frac{1}{K-1}\beta_1 \le \beta_3 - \frac{1}{K-1}\beta_1 \le \cdots \le \beta_K - \frac{1}{K-1}\beta_1$, it reduces to the assumed case, thus proving the proposition.

Now, we derive an algorithm for obtaining a reasonable noise allocation from the above induction. It follows that after $t < k - 3$ inductive iterations, the allocation for the $t$-th client is as follows:

$$
[b_{t1}, b_{t2}, \cdots, b_{t(t-1)}, 0, b_{t(t+1)}, \cdots, b_{tk}]
$$
$$
= [\theta_1, \theta_2, \cdots, \theta_{t-1}, 0, \theta_t, \cdots, \theta_t].
$$

Unlike previous inductions, we perform one additional iteration to reduce the system to 2-dimensional. By adjusting the second-largest $\beta_{k-1} \leftarrow \beta_k$, the resulting allocation satisfies the non-negativity condition, as detailed below:

$$
\begin{cases}
\theta_1 = \dfrac{1}{k-1}\beta_1 \\[3mm]
\theta_n = \dfrac{1}{k-n}\left(\beta_n - \displaystyle\sum_{i=1}^{n-1}\theta_i\right), \ 2 \le n \le k-2 \\[4mm]
\theta_{k-1} = \beta_k - \displaystyle\sum_{i=1}^{k-2}\theta_i
\end{cases}
$$

At this point, the allocation matrix for (10) is as following:

$$
\begin{bmatrix}
0 & \theta_1 & \theta_1 & \cdots & \theta_1 & \theta_1 \\
\theta_1 & 0 & \theta_2 & \cdots & \theta_2 & \theta_2 \\
\theta_1 & \theta_2 & 0 & \cdots & \theta_3 & \theta_3 \\
\vdots & \vdots & \vdots & \ddots & \vdots & \vdots \\
\theta_1 & \theta_2 & \theta_3 & \cdots & 0 & \theta_{k-1} \\
\theta_1 & \theta_2 & \theta_3 & \cdots & \theta_{k-1} & 0
\end{bmatrix}
$$

The proposition is thus proven.

$\square$

# F   ADDITIONAL EXPERIMENTS

We conducted additional experiments using the MNIST dataset.

## F.1   SCALABILITY EVALUATION

Now, we extensively explore the relationship between the client size $N$, sampling rate $q$, and privacy budget $\epsilon$. As shown in Fig.3, we traverse the sampling rate $q \in [0.1, 0.5]$ while recording the test accuracy over 200 iterations. The green lines represent the test results of FL without DP, while the blue and orange lines represent LDP-MA and NADP, respectively.

It can be observed that, the blue and orange lines intersect, indicating a $q_{same}$ at which the utility of two frameworks are the same. When the actual sampling rate $q < q_{same}$, LDP performs better; conversely, when $q > q_{same}$, NADP outperforms. As scale increases, on the one hand, the intersection point of the two lines shifts to the left, suggesting that NADP becomes more advantageous in more extensive settings. On the other hand, as the requirements for privacy protection increase (i.e., a smaller $\epsilon$) and more clients are involved, the utility advantage of NADP grows, demonstrating better scalability.

It is worth noting that in some settings where the proportion of aggregated clients is extremely small (e.g., $N = 20$, $q = 0.1$, with only 2 clients participating in aggregation), LDP outperforms NADP. This is because in IDP, we regard all clients as a collective entity rather than the sampled clients, which leads to a potentially slightly more relaxed noise bound that satisfies DP. Indeed, we could more precisely estimate the amount of data sampled each time to optimize our framework for more extreme scenarios. However, this might prevent us from explicitly comparing the noise accumulation of LDP and NADP in theory. Therefore, presenting a directly comparable result is more desirable to us, and optimization of NADP in extreme scenarios is not the focus of this paper.

## F.2   RELIABILITY EVALUATION

As a potential threat, dropout of clients results in some noise, which should have been canceled out, remaining in the model. In this subsection, we set $N = 25, 50, 100$, $q = 0.8$, with $\lambda = 1.0, 1.2, 1.4$ in NADP and $\epsilon = 1, 2$, $\delta = 10^{-5}$. We vary the dropout rate from 0.1 to 0.5 to examine the reliability.

As shown in Fig.4, FL without DP is not significantly affected by dropout, since the dropout of some clients merely results in a reduced sampling rate. However, for DP-FLs, dropout decreases the base of aggregation average, amplifying the uncertainty caused by noise, which ultimately affects overall performance. Nevertheless, experiments show that this impact is quite limited. It can be observed that the accuracy of both frameworks decreases smoothly as the dropout rate increases, indicating that both LDP and NADP exhibit a certain level of robustness against dropout. Specifically, NADP is more affected than LDP. However, due to the inherent accuracy advantage of NADP, its accuracy only falls slightly below that of LDP at high dropout rates, by which point the accuracy of both frameworks dropped to an unacceptable level. While NADP still outperforms LDP at lower dropout rates. Second, examining the intersection of the LDP and NADP curves, it is apparent that as the scale increases, the intersection shifts to the right, indicating that NADP's robustness improves relative to LDP in large-scale settings. Lastly, while introducing the noise compensation coefficient $\lambda$ impacts accuracy under dropout, this impact is also quite limited. As the scale increases, this impact is further diminished.

## F.3   PRIVACY EVALUATION

In this subsection, to demonstrate the advantage of NADP in providing independent protection for both phases, we conduct attacks on uplink phase of NADP and LDP with the same utility (i.e., same noise intensity after aggregation, same downlink protection strength, but different $\epsilon$), by employing a commonly used attack method, DLG (Zhu et al., 2019) and its variant iDLG (Zhao et al., 2020). In our experiments, we set the total number of clients $N = 100$, and all clients possess the same amount of data. sampling rate $q = 0.8$, total number of iterations $T = 200$. We examine the case where datasets are evenly distributed, and $\epsilon = 5$ and $20\sqrt{2}$ corresponding to NADP and LDP, respectively, to keep the noise intensity accumulated in the model the same. At the same time, we set the noise compensation coefficient $\lambda = 1.2$, enabling our NADP to theoretic prevent $\tau_{theoretical} = 30\%$ of

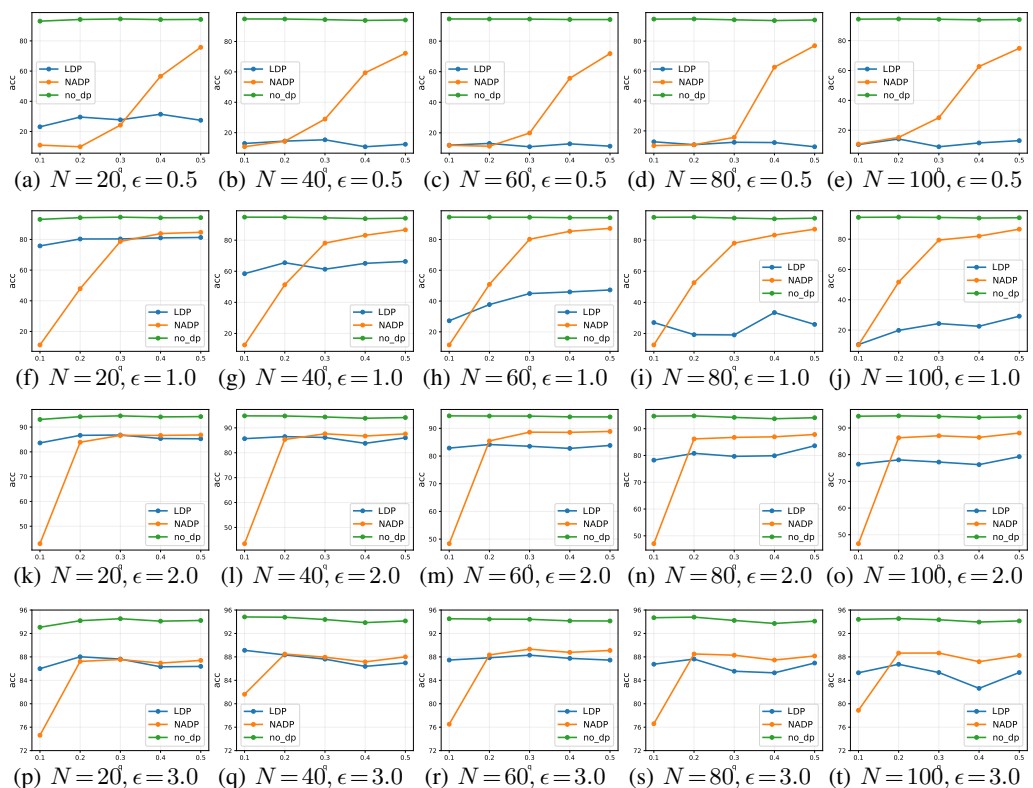

Figure 3: Performance Comparison under Different Sampling Rates on MNIST. We set the number of rounds to $T = 200$ and conduct experiments with different client size $N$ and privacy parameters $\epsilon$. We evaluate the performance of NADP and LDP across a range of sampling rates $q \in [0.1, 0.5]$.

collusion, and we traverse the practical collusion rate $\tau_{practical} \in [0.1, 0.9]$ to show the results when the preset compensation factor does not meet the actual collusion ratio.

For defense evaluation, we made the attack as powerful as possible: each client uses only one sample in local training, and the attack was conducted during the first iteration. For (i)DLG attacks, we use the L-BFGS (Tankaria et al., 2022) optimizer, performing 300 iterations to ensure convergence, and repeat the experiment 10 times to obtain the worst-case defense results. Based on this, we compared the reconstructed images with the real images, calculating the normalized Mean Squared Error (MSE), Learned Perceptual Image Patch Similarity (LPIPS) and Structural Similarity Index Measure (SSIM).

As shown in Table 4, with upward and downward arrows indicating the direction of better defense performance, the most prominent reconstructed images are placed in the last row to demonstrate whether they can be recognized by the human eye. It can be observed that, without DP, the (i)DLG attacks can almost perfectly reconstruct the original images. After applying DP, the reconstructed images are significantly disrupted, and, NADP exhibits a more effective defense. Additionally, the observation aligns with the earlier theoretical findings: the residual noise in NADP follows the same cumulative pattern as in LDP. As shown in Table 4, when the practical collusion rate $\tau \to 1$, that is, when all remaining clients collude (which is almost impossible in real-world scenarios), the risk of privacy leakage facing the client is the same as in LDP. However, counteracting noise provides additional privacy protection, amplified quadratically by the compensation coefficient $\lambda > 1$, with no loss in accuracy. This shows that NADP can provide more effective privacy protection with lower utility loss.

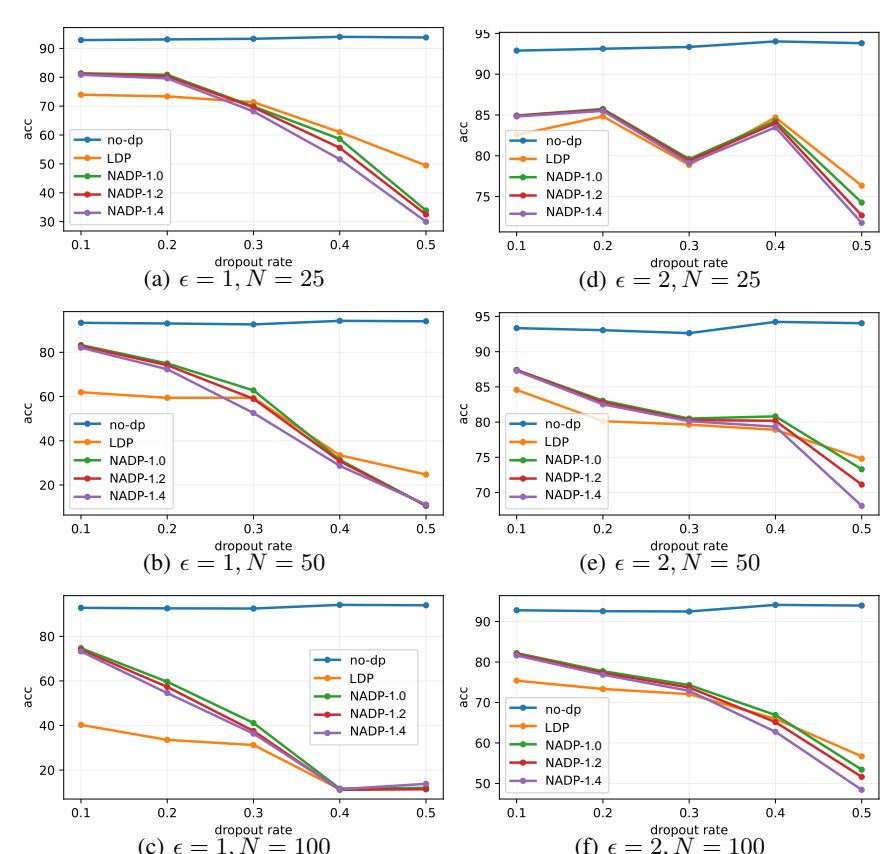

Figure 4: Performance Comparison under Different Dropout Rates on MNIST. We conduct experiments with different $N$ and privacy budget $\epsilon = 1, 2$. We evaluate the performance of NADP ($\lambda = 1.0, 1.2, 1.4$), LDP and FL without DP across a range of dropout rates $d \in [0.1, 0.5]$ over 200 iterations.

Table 4: MSE, LPIPS, SSIM results under DLG/iDLG Attacks

| | Framework | FL(no-dp) | LDP($\epsilon = 20\sqrt{2}$) | NADP($\epsilon = 5, \lambda = 1.2$) | | | | |
| | | | | $\tau = 0.9$ | $\tau = 0.7$ | $\tau = 0.5$ | $\tau = 0.3$ | $\tau = 0.1$ |
|---|---|---|---|---|---|---|---|---|
| MNIST | MSE ↑ | 3.48e-4 / 1.89e-4 | 3.33e-3 / 2.78e-3 | 0.0291 / 0.0251 | 0.0331 / 0.0265 | 0.0768 / 0.0536 | 0.141 / 0.149 | 0.224 / 0.212 |
| | LPIPS ↑ | 4.81e-5 / 4.63e-5 | 1.55e-2 / 1.23e-2 | 0.0663 / 0.0737 | 0.0821 / 0.0847 | 0.105 / 0.115 | 0.188 / 0.194 | 0.355 / 0.353 |
| | SSIM ↓ | 0.9390 / 0.9439 | 0.8228 / 0.8186 | 0.6527 / 0.6840 | 0.6665 / 0.6841 | 0.6165 / 0.6488 | 0.5287 / 0.5201 | 0.2900 / 0.2922 |
| | | 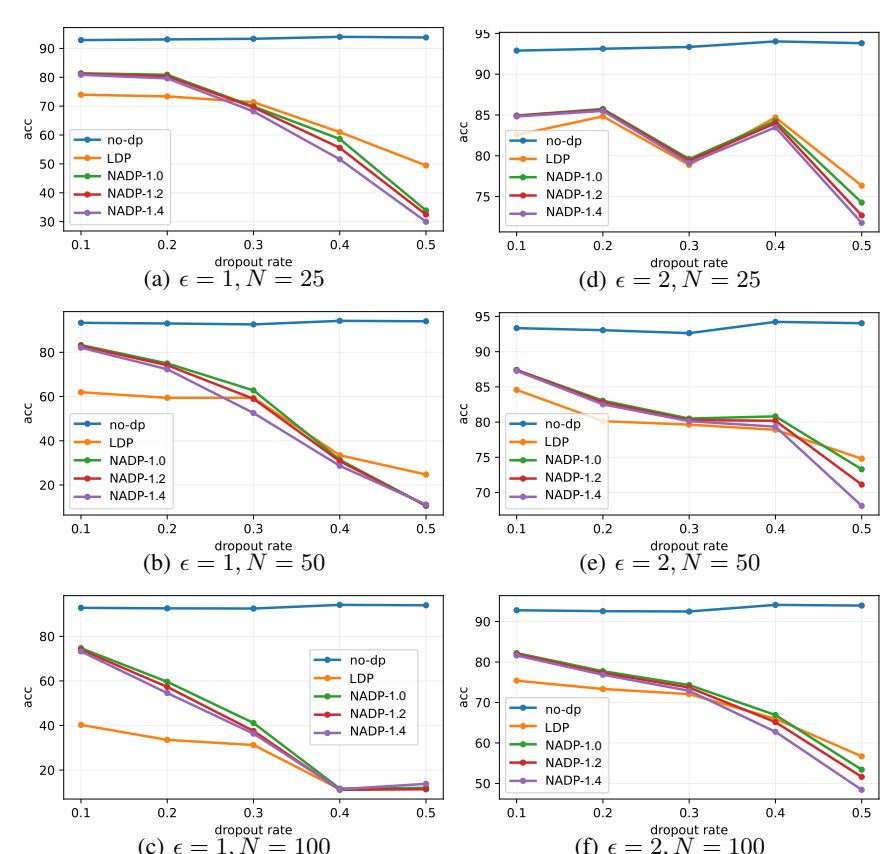 | | | | | | |
| Fashion-MNIST | MSE ↑ | 7.58e-5 / 4.24e-5 | 1.49e-3 / 3.36e-3 | 0.0252 / 0.0178 | 0.0379 / 0.0229 | 0.0683 / 0.0534 | 0.132 / 0.0846 | 0.0939 / 0.119 |
| | LPIPS ↑ | 7.22e-6 / 6.96e-6 | 3.61e-3 / 3.70e-3 | 0.0453 / 0.0359 | 0.0983 / 0.0579 | 0.155 / 0.172 | 0.199 / 0.186 | 0.297 / 0.315 |
| | SSIM ↓ | 0.9575 / 0.9463 | 0.8704 / 0.8482 | 0.7757 / 0.7916 | 0.7202 / 0.7238 | 0.5714 / 0.5642 | 0.5437 / 0.5221 | 0.3811 / 0.3824 |
| FEMNIST | MSE ↑ | 3.54e-4 / 2.67e-4 | 3.84e-3 / 5.94e-3 | 0.0171 / 0.0163 | 0.0756 / 0.0577 | 0.0468 / 0.0821 | 0.0718 / 0.0871 | 0.117 / 0.105 |
| | LPIPS ↑ | 2.41e-5 / 3.24e-5 | 7.56e-3 / 4.05e-3 | 0.0886 / 0.0774 | 0.0811 / 0.0939 | 0.0985 / 0.1092 | 0.179 / 0.213 | 0.257 / 0.318 |
| | SSIM ↓ | 0.9875 / 0.9647 | 0.8817 / 0.9105 | 0.8444 / 0.8135 | 0.8220 / 0.7669 | 0.8164 / 0.7250 | 0.6842 / 0.6703 | 0.5994 / 0.5060 |

