# OpenReview forum: "Achieving Better Utility beyond LDP-FL by Independent Two-phase Protection"
_ICLR.cc/2026/Conference — ICLR 2026 Conference Withdrawn Submission_

### Official Review · Reviewer_Jer2 · 2025-10-22

**Soundness:** 2
**Presentation:** 2
**Contribution:** 2
**Rating:** 2
**Confidence:** 4

**Summary:**

The paper revisits Local Differential Privacy Federated Learning (LDP-FL) and claims that its sequential design causes “noise redundancy,” where uplink noise unnecessarily propagates to the downlink phase and degrades model utility. To address this, the authors propose an idealized framework, IDP-FL, with independent privacy protection for uplink and downlink, and a practical instantiation, NADP-FL, that uses pairwise “noise annihilation” among clients to cancel redundant noise while preserving differential privacy. Experiments on MNIST, Fashion-MNIST, FEMNIST, and IMDb show improved accuracy compared to standard LDP-FL.

**Strengths:**

- Identifies a structural inefficiency (noise redundancy) in LDP-FL.

- Introduces an innovative idea of paired-noise annihilation.

- Shows promising empirical improvements in controlled experiments.

**Weaknesses:**

- The clarification of "downlink privacy requirements" is not clear. It seems that the downlink shares the same requirement as the uplink; thus, due to the noise amplification effect, the \epsilon will be smaller compared to the client side. This noise amplification effect has been studied before.
- NADP-FL assumes all clients are honest and properly generate shared random seeds, although some malicious scenarios are analyzed. However, how to produce paired noise seems to be unclear. In addition, the communication overhead of the Diffie-Hellman protocol is missing.
- During the noise aggregation, it seems that it follows the standard Fed-AVG algorithm; the reviewer may wonder if other robust/advanced aggregation schemes can support the proposed algorithm.
- Comparisons with other DP-based schemes are missing, i.e., Adaptive DP or Shuffle-DP, which also optimize utility-noise tradeoffs.

**Questions:**

- Does the scheme work with robust/advanced aggregation schemes?
- Why do all experiments assume IID data with |D_max|/|D_min|<=2?
- What is the overhead of the Diffie-Hellman protocol? The reviewer may think it could be O(n^2) for generating pairwise keys, and it could be a disadvantage with a large number of clients.

---

### Official Review · Reviewer_nkwZ · 2025-10-29

**Soundness:** 2
**Presentation:** 2
**Contribution:** 2
**Rating:** 2
**Confidence:** 3

**Summary:**

While the paper identifies an interesting potential limitation in LDP-FL, termed "noise redundancy," the proposed solution (NADP-FL) is fundamentally flawed in its framing, evaluation, and technical contribution. The core mechanism is a reimplementation of a simplified Secure Aggregation (SA) + DP paradigm, which it then compares unfairly to both LDP and more robust SA protocols.

**Strengths:**

+ The paper provides a clear and well-articulated motivation by identifying "noise redundancy" in LDP-FL, where the noise required for uplink privacy may be excessive for the downlink phase.

+ The conceptual idea of "noise annihilation" , where clients generate counteracting noise pairs that cancel during aggregation, is an intuitive approach to improving the privacy-utility trade-off.

**Weaknesses:**

- The core mechanism, which relies on clients using the Diffie-Hellman protocol to generate paired noises that cancel during aggregation, is functionally a specific implementation of the well-known SA + DP paradigm. While the authors frame this as an improvement "beyond LDP-FL", it fundamentally changes the LDP threat model by introducing cryptographic assumptions and inter-client key exchange. This positions the work directly against existing SA+DP literature, rather than as a novel improvement to LDP itself. The central idea is to use SA to reduce the noise required for DP, which is not a new contribution.


- The overhead comparison to SA-FL (Bonawitz et al., 2017)  is unconvincing. The proposed NADP-FL protocol appears to be a simplified SA mechanism that does not account for key features, such as robustness to client dropouts. The paper's own experiments (Appendix F.2) confirm that dropouts negatively impact NADP, and potentially more so than LDP. Therefore, claiming lower overhead  than a more feature-rich and robust protocol is an unfair comparison and does not provide a meaningful assessment of the protocol's efficiency.

- The paper lacks a formal, provable security analysis for its cryptographic protocol. While it provides a privacy analysis to demonstrate $(\epsilon, \delta)$-DP, it does not offer a formal security proof (e.g., via simulation) for the "noise annihilation" mechanism itself, especially in the presence of malicious clients. The discussion in Appendix A 12 is informal and insufficient. This stands in contrast to foundational SA works, which provide rigorous, provable security guarantees.

- The experimental results for utility are not compelling, especially when viewing the framework as an SA+DP solution. The reported accuracies on MNIST (e.g., ~80-87% for N=100 ) are significantly lower than what is often achieved by standard SA+DP implementations, which can surpass 90% accuracy even with strong privacy guarantees. This suggests that the proposed method does not, in practice, deliver a superior privacy-utility trade-off compared to existing, more robust solutions.

**Questions:**

See Weakness

---

### Official Review · Reviewer_dR2V · 2025-11-01

**Soundness:** 2
**Presentation:** 2
**Contribution:** 2
**Rating:** 2
**Confidence:** 2

**Summary:**

The authors propose an ideal interaction mode, Ideal Differential Privacy Federated Learning (IDP-FL), which decouples the uplink and downlink phases for independent privacy protection. They then introduce Noise Annihilation Differential Privacy Federated Learning (NADP-FL) as a practical instantiation of IDP-FL, where clients generate paired noises that mutually cancel during aggregation, eliminating redundancy while maintaining (ϵ, δ)-DP guarantees. Theoretical analysis proves that NADP-FL requires less noise in the downlink than LDP-FL under equivalent privacy budgets. Experiments validate improved utility, scalability with client count, and robustness to client dropouts compared to LDP-FL baselines.

However, it would be valuable to formally examine scenarios where a client deliberately sends malformed or adversarial noise to disrupt the annihilation process, supported by either a formal security proof or an empirical evaluation under such an attack model. In addition, the theoretical complexity analysis should be complemented with empirical overhead measurements, reporting wall-clock time for the one-time key setup and per-round client computation (including noise generation and allocation) as the number of clients N and key dimensions k increase, to better illustrate the framework’s practicality.

**Strengths:**

The paper uncovers a fundamental structural defect in LDP-FL (noise redundancy) through rigorous theoretical analysis and proofs—a novel insight that advances understanding of privacy-utility trade-offs in DP-FL.

**Weaknesses:**

--The overall implementation relies on a semi-honest model, which, although similar to local differential privacy, is still an overly strong assumption.
The core framework and its analysis assume a semi-honest setting where clients and the server "will follow the protocol exactly". This is a strong assumption. The discussion of malicious scenarios (where clients might deviate from the protocol, e.g., by not generating noise correctly or skipping the key exchange) is limited and relegated to an appendix. The claim that the framework "will not collapse" is not a formal security guarantee.
--The practical overhead issue of this method.
The NADP-FL framework introduces significant overhead not present in LDP-FL. This includes a "Preparation Phase" for all clients to perform pairwise Diffie-Hellman key exchange and, in each round, the local computation of k−1 paired noise sequences and the execution of the complex noise allocation algorithm (Alg. 2). While the complexity is analyzed (Table 2), the practical scalability of this overhead (e.g., key management in a dynamic FL system with thousands of clients) is a major concern.
--This method is quite sensitive to user disconnections.
The "noise annihilation" mechanism fundamentally relies on the successful aggregation of all k paired noises. The paper acknowledges that client dropout leads to "incomplete noise cancellation". While experiments in Appendix F.2 show the impact is "limited" , they also indicate that NADP's accuracy is "more affected than LDP" by dropout. This introduces a new fragility, as dropout is common in real-world federated systems.
--The noise allocation in this method is relatively complex and may be difficult to implement in practice.
The proposed noise intensity allocation (Algorithm 2) is non-trivial. It requires sorting clients by dataset size and solving a system of equations. The paper itself notes that "finding a non-negative solution becomes challenging" in some cases and resorts to an "approximate solution". This complexity adds computational burden on the clients and may be a barrier to practical implementation.

**Questions:**

Please see weaknesses.

---

### Official Review · Reviewer_xaNp · 2025-11-03

**Soundness:** 2
**Presentation:** 3
**Contribution:** 2
**Rating:** 4
**Confidence:** 4

**Summary:**

The paper revisits Local Differential Privacy in Federated Learning (LDP-FL), identifying an inherent flaw called noise redundancy, where uplink noise unnecessarily propagates into the downlink phase. The authors propose an idealized framework, LDP-FL, allowing independent protection in both phases, and instantiate it as NADP-FL, which uses structured noise pairs that cancel during aggregation. Extensive experiments on MNIST, Fashion-MNIST, FEMNIST, and IMDb and plus additional tests on scalability, dropout, and DLG/iDLG privacy leakage which show that NADP-FL improves utility under strong privacy settings.

**Strengths:**

- **S1:** The proposed framework highlights a structural inefficiency (noise redundancy) in standard LDP-FL.
- **S2:** The authors propose a theoretically motivated fix (NADP-FL) with formal DP guarantees.
- **S3:** Extensive experiments across datasets and scenarios, including DLG/iDLG leakage tests demonstrate strong scalability and utility at tighter privacy budgets.

**Weaknesses:**

- **W1:** The idea of decoupling uplink and downlink privacy is conceptually novel but incrementally extends prior work on adaptive or composition-aware DP. The “noise annihilation” mechanism is elegant but operationally unrealistic due to O(n²) pairwise coordination. The work provides useful theoretical framing but limited practical innovation.
- **W2:** The theoretical arguments are internally consistent but depend on unrealistic assumptions such as perfect synchronization, honest participation, and stable communication. The privacy proofs are correct but idealized, and scalability analysis remains simulation-based. While NADP-FL clearly improves performance over LDP-FL in controlled settings, the practicality and generality of the approach are questionable.
- **W3:** The paper is mathematically heavy and difficult to follow. Algorithms are notation-heavy and poorly explained. Figures are dense, and several discussions (e.g., Appendix F) repeat earlier content. The paper’s structure could be much clearer, with more intuition and less algebraic formalism.
- **W4:** Lacks comparison with DP-SGD, Secure Aggregation, or Shuffle-DP.

**Questions:**

- **Q1:** How does NADP-FL behave under asynchronous or partial participation (e.g., clients joining or dropping mid-round)
- **Q2:** What are the concrete communication and computation overheads (in time, bandwidth, or FLOPs) compared to LDP-FL and Secure Aggregation?
- **Q3:** How does NADP interact with encryption-based aggregation? Would annihilation still occur under secure aggregation protocols?
- **Q4:** Are there potential correlations between client noise pairs that could leak structure or enable reconstruction under adversarial conditions?
- **Q5:** Could adaptive or data-dependent noise scaling achieve similar benefits without pairwise coordination? How would NADP handle non-IID data distributions, which are typical in FL and affect both variance and privacy composition?

---

### Note · Authors · 2025-11-13

I have read and agree with the venue's withdrawal policy on behalf of myself and my co-authors.